**Research**

# Is increasing urbanicity associated with changes in breastfeeding duration in rural India? An analysis of cross-sectional household data from the Andhra Pradesh children and parents study

Laura Oakley,[1] Christopher P Baker,[1] Srivalli Addanki,[2] Vipin Gupta,[3] Gagandeep Kaur Walia,[4] Aastha Aggarwal,[4] Santhi Bhogadi,[2] Bharati Kulkarni,[5] Robin T Wilson,[6] Dorairaj Prabhakaran,[4] Yoav Ben-Shlomo,[7] George Davey Smith,[7] K V Radha Krishna,[5] Sanjay Kinra[1]

For numbered affiliations see end of article.

**Correspondence to**
Dr Laura Oakley;
laura.oakley@lshtm.ac.uk

## ABSTRACT

**Objective** To investigate whether village-level urbanicity and lower level socioeconomic factors are associated with breastfeeding practices in transitioning rural communities in India.

**Setting** 29 villages in Ranga Reddy district, southern India between 2011 and 2014.

**Participants** 7848 children under 6 years identified via a cross-sectional household survey conducted as part of the Andhra Pradesh Children and Parents Study.

**Outcome measures** Two key indicators of optimal breastfeeding: termination of exclusive breastfeeding before 6 months and discontinuation of breastfeeding by 24 months. Village urbanicity was classified as low, medium or high according to satellite assessed night-light intensity.

**Results** Breastfeeding initiation was almost universal, and approximately two in three children were exclusively breastfed to 6 months and a similar proportion breastfed to 24 months. Using multilevel logistic regression, increasing urbanicity was associated with breastfeeding discontinuation before 24 months (medium urbanicity OR 1.45, 95% CI 0.71 to 2.96; high urbanicity OR 2.96, 95% CI 1.45 to 6.05) but not with early (<6 months) termination of exclusive breastfeeding. Increased maternal education was independently associated with both measures of suboptimal breastfeeding, and higher household socioeconomic position was associated with early termination of exclusive breastfeeding.

**Conclusion** In this transitional Indian rural community, early stage urbanicity was associated with a shorter duration of breastfeeding. Closer surveillance of changes in breastfeeding practices alongside appropriate intervention strategies are recommended for emerging economies.

## Strengths and limitations of this study

► Previous studies have investigated the association between urbanisation and breastfeeding using the urban–rural dichotomy.

► We used data from a large rural cohort in southern India that is currently undergoing rapid and uneven urbanisation due to its proximity to a major urban centre.

► The use of night-time light intensity data as an indicator of urbanicity allowed us to examine subtler changes in breastfeeding practices along the urban–rural continuum.

► Sixteen per cent of children were excluded from the analysis due to missing information on breastfeeding practices.

► We relied on maternal retrospective recall of breastfeeding events for our outcome measurement.

death.[1] Optimal breastfeeding is defined by the WHO as early breastfeeding initiation, exclusive breastfeeding (EBF) to 6 months and continued breastfeeding to 2 years or beyond alongside appropriate complementary feeding. Many low-income and middle-income countries (LMICs) have a strong tradition of near universal and prolonged breastfeeding,[2 3] though EBF to 6 months (hereafter referred to simply as 'exclusive breastfeeding') is less common. A small increase in the global proportion of children exclusively breastfed between 1995 and 2010 has been reported, but the overall proportion (40%) still falls strikingly short of universal coverage and obscures differences in country-specific trends.[4]

## INTRODUCTION

The promotion of breastfeeding is one of the three interventions identified as having the largest potential impact on global child

**BMJ**

Many LMICs are currently experiencing a rapid increase in the proportion of people living in built-up areas, and the social, cultural and economic changes associated with this process of urbanisation have the potential to impact on traditional breastfeeding practices. Direct threats to optimal breastfeeding include early introduction of other liquids and inappropriate supplementation with solid or semisolid foods. These behaviours may be influenced by changing social norms, for example, increasing numbers of mothers working outside the home. Of all positive health behaviours, breastfeeding is one of the few more prevalent in LMICs compared with high-income countries.[3] Within LMICs, this trend is mirrored by a higher prevalence of suboptimal breastfeeding in urban areas compared with rural areas[5]: a trend also observed in India[6–8] alongside variation by various socioeconomic indicators.[6–10] Although the high-level urban–rural comparison is of interest, there may also be subtler changes in breastfeeding practices along the urban–rural continuum given the periurban effects on villages close to urban centres. These changes can potentially be investigated by using a measure of 'urbanicity' that aims to assess the extent of urbanisation in a given area. A number of different indicators of urbanicity have evolved, including the use of remote light sensing[11 12] and multicomponent scales.[13 14] The early identification of changes in breastfeeding practices accompanying the urbanicity transition—and an understanding of the underlying mechanisms—are necessary for informing appropriate interventions to protect traditionally positive breastfeeding practices in transitioning communities.

The Andhra Pradesh Children and Parents Study (APCAPS) is a rural sociodemographic cohort in southern India that is currently undergoing rapid and uneven urbanisation due to its proximity to a major urban centre (Hyderabad), providing a unique opportunity to examine the association between early stage urbanicity and breastfeeding practices.

## METHODS
### Study design
APCAPS is an intergenerational cohort originally established to study the long-term effects of early-life undernutrition on risk of cardiovascular disease and subsequently expanded to include transgenerational influences of other environmental and genetic factors on chronic diseases in transitioning rural India.

The original cohort is based on the participants in the Hyderabad Nutrition Trial conducted in 1987–1990 in 29 villages approximately 50–100 km from Hyderabad in Telangana state (formally Andhra Pradesh), southern India.[15] The dataset used in this analysis is based on a cross-sectional household survey conducted between 2011 and 2014 in the study villages. All households (household defined as a group of people living in the same residence and sharing a common kitchen) in the study villages were visited by fieldworkers, and sociodemographic information was collected on each household. In addition, a basic health profile was collected for each child under 6 years of age, comprising information on infant feeding (colostrum intake, total duration of breastfeeding and age of onset of weaning), immunisation and anthropometric measurements. Fieldworkers made repeated visits to households to maximise response and to clarify inconsistencies in collected data. Data were collected from 23 314 households in total, of which 5968 (25.6%) included at least one child under 6 years.

The study received approval from the ethics committees of the National Institute of Nutrition (Hyderabad, India) and London School of Hygiene and Tropical Medicine (London, UK). Approval was also sought from the Indian Council for Medical Research and the village heads and their committees in each of the study villages. Written informed consent (or witnessed thumbprint if illiterate) was obtained from the participants prior to their inclusion in the study.

### Breastfeeding outcomes and explanatory variables
Two breastfeeding outcomes were used in this analysis: termination of EBF before 6 months and discontinuation of breastfeeding before 24 months. These outcomes reflect failure to achieve two of the specific WHO recommendations for optimum feeding practices (EBF to 6 months and continued breastfeeding to 2 years).[16] As part of the basic health profile for children compiled for children under 6 in the household survey, mothers were asked to report the total duration of breastfeeding (in months), and the age (in months) at onset of weaning. 'Weaning' was defined by fieldworkers as the age at which the child was given anything other than mother's milk, that is, age at initiation of complementary feeding. A copy of the questions used in the survey is provided as a supplementary figure (see online supplementary figure S1).

Our primary explanatory factor was urbanicity, measured using remotely sensed village-level night-time light intensity (NTLI) scores, as these are objective, unbiased and easily available over wide areas. Although this analysis represents the first application of NTLI data to the APCAPS population, NTLI data are increasingly being used as an area-based indicator of socioeconomic development.[11 12] The light that is included in the NTLI score include any outside lights, ranging from fires and gas flares to lights related to human settlements. Low-level lights such as from streets and car headlights can be observed if there is a sufficient number of sources, but indoor lights cannot be observed. NTLI scores were calculated for 2012 using the National Oceanic and Atmospheric Administration Stable Lights product that provides yearly average NTLI measures processed and filtered to remove events such as fires and lightening contamination by cloud or moon reflections and background noise, at a 1 km resolution. Scores for each village were calculated by summing the raw NTLI values over each village polygon (digitised using Bing Maps combined with GPS-based surveying by the field teams). The 1 km resolution NTLI data were

upscaled to 100 m resolution to allow more accurate estimation of the NTLI values covered by each polygon, as many villages are small and partially cover multiple 1 km grid cells.

NTLI scores for each village were validated against alternative urbanicity measurements (field worker ranking and a multicomponent urbanicity score based on household-level material assets and village-level availability of infrastructure and services) showing positive correlations (0.65 and 0.53, respectively). Study villages were ranked by their NTLI score and divided into tertiles to represent 'low'' (10 villages), 'medium' (10 villages) and 'high' (nine villages) levels of urbanicity. The NTLI tertile scores matched the field worker ranking in 50% of the villages, and cases of disagreement between NTLI and field worker ranking, the latter was more conservative and ranked villages as medium urbanicity rather than high urbanicity. Only one village had a significant divergence between NTLI and fieldworker ranking.

In addition, we investigated mother-level socioeconomic factors that may be correlated with urbanicity: maternal education (no formal education, primary education, or secondary education and higher), maternal employment (paid work vs no paid work) and a household level standard of living index (SLI). Asset-based SLIs have been established as a valid proxy measure of household wealth.[17] We generated an SLI score for each household, calculated by using information on household assets including house and land ownership, characteristics of the home (electricity, water pump, separate kitchen and separate toilet) and ownership of various assets (tractor, radio, AC, washing machine, bore hole, telephone, TV, fridge, bicycle, two wheeler, four wheeler, bank account, animal cart, sofa, cot/bed, mattress and table). Principal component analysis was used to determine the weights for each component in the index,[18] and households were divided into quintiles according to their weighted score. We also report data on a number of other factors likely to be associated with breastfeeding practices: sex of child, birth order, maternal age (grouped) and household composition (joint/extended or nuclear).

## Statistical analysis

We included in the analysis all children under 6 years who were breastfed at least once and for whom information was available on feeding history. The analysis investigating termination of EBF before 6 months was restricted to children 6 months or older at the time of survey, and correspondingly only those children aged 24 months or older were included in the analysis of discontinuation of breastfeeding before 24 months. A small proportion of children (3%) had missing information on one or more variables of interest and were excluded.

We hypothesised that urbanicity would be associated with less favourable breastfeeding practices. This could operate through at least two different indirect pathways (see figure 1): through increasing individual-level employment, education or assets so that households

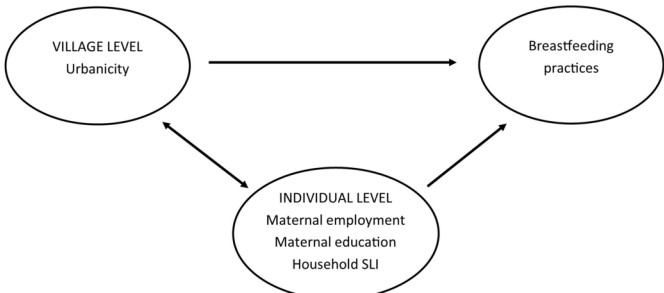

**Figure 1** Model of the association between village level urbanicity, individual level socioeconomic indicators and breastfeeding practices.

are less likely to maintain breastfeeding, or due to more urbanised villages have a different 'collective' attitude to breastfeeding. We investigated these hypotheses by using multilevel logistic regression modelling with children (level 1) nested within mothers (level 2, max n=5477) nested within villages (level 3, n=29). This approach allowed us to model the variation in breastfeeding outcomes at each level (random effects) and to estimate the effect of specific mother and village-level factors on breastfeeding practices (fixed effects). We initially fitted a null model (model 1) for each of the two outcomes with random intercepts only in order to estimate the baseline between-mother and between-village variance. We then fitted a series of models for each breastfeeding outcome, adding covariates as fixed effects to the included random effects, where fixed effects were interpreted as the average effect on the specified breastfeeding outcome across all mothers and villages. These models included individual demographic factors and mother-level socioeconomic indicators (model 2), individual demographic factors and village-level urbanicity (model 3) and all variables (model 4). Due to the correlation between socioeconomic indicators and urbanicity, we considered estimates from model three our main results. Proportional change in variance was calculated as a measure of change in mother-level (level 2) and village-level (level 3) variance between the null model and subsequent models, and (for village-level variance only) the measure of change between a model with (model 4) and without (model 2) the village-level urbanicity variable included.

Estimates of the association between mother-level socioeconomic variables and breastfeeding outcomes were derived from model 2 (adjusted for individual-level demographic variables, but not urbanicity).

We hypothesised a priori that the association between village-level urbanicity and breastfeeding may vary by household SLI and maternal education. We investigated these cross-level interactions in further models (for SLI, comparing the richest two quintiles to the three poorest quintiles; for education, comparing secondary education vs no or primary education).

All statistical analyses were conducted using Stata version14.

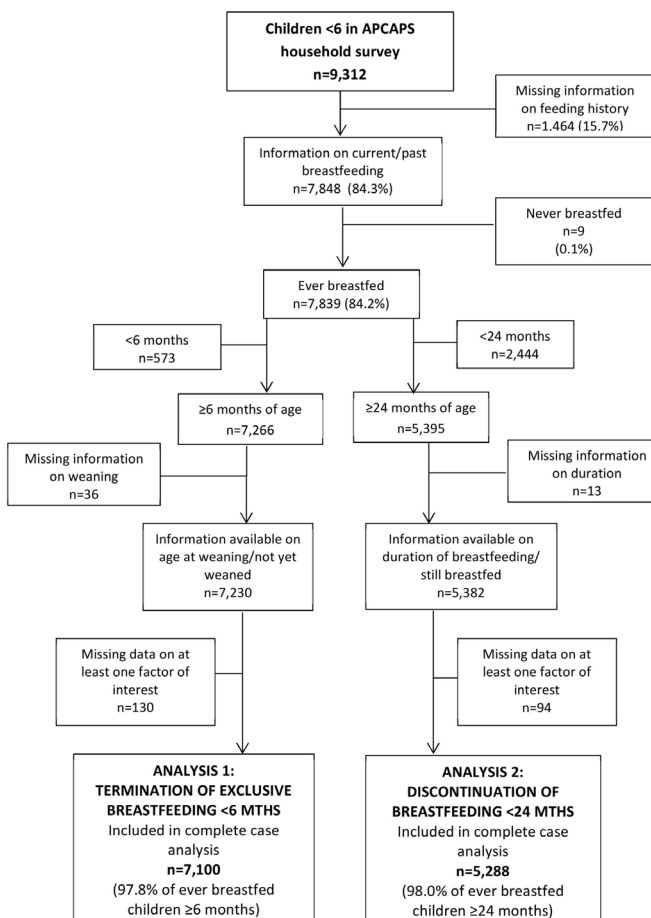

**Figure 2** Flow chart of how samples for the two breastfeeding indicators were reached.

## RESULTS

### Characteristics of the sample

Information on breastfeeding was available on a total of 7848 children (5390 households), 99% (n=7839) of whom were breastfed at least once (figure 2).

The characteristics of ever-breastfed children by urbanicity of village are presented in table 1. There was little variation in infant sex, birth order or maternal age by urbanicity of village. Children residing in villages classified as more urbanised had mothers that were more likely to have been educated to secondary level, less likely to have mothers in paid employment and a higher standard of living. Joint/extended families were slightly less prevalent in high urbanicity villages.

### Termination of exclusive breastfeeding by 6 months

Among the 7142 children no longer exclusively breastfed (88 children ≥6 months were still exclusively breastfed at the time of survey), the mean age at termination of EBF was 6.1 months (SD 1.8), median 6.0 months and IQR 5–6 months. One-third of children (33.5%, n=2420) were EBF for a period of less than 6 months (table 1).

### Fixed effects

There was no statistically significant trend regarding early termination of EBF and village level urbanicity.

The prevalence of early termination of EBF was lowest in medium urbanicity villages (27.2%), higher in lower urbanicity villages (33.6%) and highest in high urbanicity villages (36.5%). In multivariable analysis, there was no evidence that urbanicity was associated with termination of EBF by 6 months (model 3, table 2), with little change in estimates after the addition of demographic and socio-economic covariates to the model.

After adjustment for other individual-level and mother-level covariates, both children of mothers with primary education and children of mothers with secondary education were more likely to be EBF for less than 6 months when compared with children of mothers with no formal education (primary education: OR 3.37, 95% CI 2.13 to 5.31; secondary education: OR 1.69, 95% CI 1.12 to 2.54; model 2, table 2). Increasing SLI quintile was associated with up to twice the odds of early termination of EBF compared with children from the poorest households (richest quintile OR 2.11, 95% CI 1.22 to 3.63; p value for trend=0.003). There was some evidence that maternal employment was also associated with early termination of EBF (OR 1.43, 95% CI 1.00 to 2.03). The estimates for socioeconomic variables did not change with the addition of urbanicity to the model.

### Interaction effects

There was no evidence of interaction between urbanicity and either household SLI or maternal education.

### Random effects

There was statistically significant unexplained variance estimates at both the mother and village level (model 1, table 1). Unexplained variability was consistently higher at the mother level compared with the village level. The addition of individual level parameters resulted in a slight decline in community-level variation (variance 1.50 and 1.44 in models 1 and 2, respectively). There was a further decline in village-level variance when the urbanicity variable was added to the model (variance 1.32 in model 4). Comparing the village-level variance between model 2 and model 4 suggests that 8.5% of the observed village-level variation can be explained by urbanicity.

### Discontinuation of breastfeeding by 24 months

At the time of survey, 784 children aged ≥24 months were still being breastfed. Among those children no longer breastfed, the mean and median age at discontinuation of breastfeeding was 21.4 (SD 8.5) and 24 months, respectively, and the IQR was 15–24 months. Nearly 4 in 10 children (37.8%, n=2037) were breastfed for less than 24 months in total.

### Fixed effects

Discontinuation of breastfeeding by 24 months was more common in high urbanicity villages (42.1%) and least common in low urbanicity villages (29.6%). After adjustment for individual-level demographic factors, high urbanicity was associated with increased odds of breastfeeding discontinuation before 24 months (OR 2.64, 95% CI 1.29

**Table 1** Characteristics of ever-breastfed children under six in the APCAPS household survey, by village urbanicity tertile (n=7839)

| | | Urbanicity (measured by night-time light intensity) | | | | | | All | |
| | | High tertile 1 (n=4276) | | Medium tertile 2 (n=2082) | | Low tertile 3 (n=1476) | | | |
| | | n | (%) | n | (%) | n | (%) | n | (%) |
|---|---|---|---|---|---|---|---|---|---|
| Exclusive breastfeeding ≥6 months* | Yes | 2495 | (63.5) | 1400 | (72.8) | 915 | (66.4) | 4810 | (66.5) |
| | No | 1436 | (36.5) | 522 | (27.2) | 462 | (33.6) | 2420 | (33.5) |
| Continued breastfeeding ≥24 months† | Yes | 1701 | (57.9) | 927 | (65.0) | 717 | (70.4) | 3345 | (62.2) |
| | No | 1236 | (42.1) | 499 | (35.0) | 302 | (29.6) | 2037 | (37.8) |
| Infant sex | Male | 2189 | (51.2) | 1069 | (51.2) | 748 | (50.5) | 4006 | (51.1) |
| | Female | 2087 | (48.8) | 1013 | (48.7) | 733 | (49.5) | 3833 | (48.9) |
| Age of child at survey | 0–1 | 1337 | (31.3) | 650 | (31.2) | 457 | (30.9) | 2444 | (31.2) |
| | 2–3 | 1520 | (35.5) | 718 | (34.5) | 535 | (36.1) | 2773 | (35.4) |
| | 4–5 | 1419 | (33.2) | 714 | (34.3) | 489 | (33.0) | 2622 | (33.4) |
| Birth order | 1 | 1824 | (43.1) | 883 | (42.7) | 607 | (41.0) | 3314 | (42.6) |
| | 2 | 1667 | (39.4) | 798 | (38.6) | 588 | (39.8) | 3053 | (39.2) |
| | ≥3 | 743 | (17.5) | 389 | (18.8) | 284 | (19.2) | 1416 | (18.2) |
| | missing | 42 | | 12 | | 2 | | 56 | |
| | mean (SD) | 1.79 | (0.85) | 1.80 | (0.85) | 1.82 | (0.85) | 1.80 | (0.85) |
| Age of mother at birth | <20 | 737 | (17.3) | 364 | (17.6) | 276 | (18.6) | 1377 | (17.6) |
| | 20–24 | 2410 | (56.5) | 1238 | (59.6) | 862 | (57.7) | 4510 | (57.7) |
| | 25–29 | 917 | (22.1) | 393 | (19.0) | 288 | (20.2) | 1598 | (20.2) |
| | 30+ | 198 | (4.6) | 78 | (3.8) | 54 | (3.6) | 330 | (4.2) |
| | missing | 14 | | 9 | | 1 | | 24 | |
| | mean (SD) | 22.9 | (3.6) | 22.7 | (3.4) | 22.7 | (3.5) | 22.8 | (3.5) |
| Family structure | Nuclear | 2866 | (67.6) | 1306 | (63.7) | 942 | (64.0) | 5114 | (65.9) |
| | Joint/extended | 1371 | (32.4) | 745 | (36.3) | 530 | (36.0) | 2646 | (34.1) |
| | missing | 39 | | 31 | | 9 | | 79 | |
| Maternal education | No formal schooling | 1054 | (24.7) | 663 | (32.0) | 558 | (37.7) | 2275 | (29.1) |
| | Primary | 880 | (20.6) | 390 | (18.8) | 284 | (19.2) | 1554 | (19.9) |
| | Secondary+ | 2329 | (54.6) | 1022 | (49.3) | 638 | (43.1) | 3989 | (51.0) |
| | missing | 13 | | 7 | | 1 | | 21 | |
| Maternal employment | Not working | 3144 | (73.8) | 1342 | (64.7) | 815 | (55.0) | 5301 | (67.8) |
| | Working | 1119 | (26.2) | 733 | (35.3) | 666 | (45.0) | 2518 | (32.2) |
| | missing | 13 | | 7 | | 0 | | 20 | |

Continued

**Table 1** Continued

| Standard of living (SLI) index | Urbanicity (measured by night-time light intensity) | | | | | | | |
|---|---|---|---|---|---|---|---|---|
| | High tertile 1 (n=4276) | | Medium tertile 2 (n=2082) | | Low tertile 3 (n=1476) | | All | |
| | n | (%) | n | (%) | n | (%) | n | (%) |
| Poorest | 611 | (14.3) | 318 | (15.3) | 242 | (16.3) | 1171 | (14.9) |
| Poorer | 736 | (17.2) | 422 | (20.3) | 288 | (19.4) | 1446 | (18.5) |
| Middle | 816 | (19.1) | 458 | (22.0) | 378 | (25.5) | 1652 | (21.1) |
| Richer | 964 | (22.5) | 497 | (23.9) | 314 | (21.2) | 1775 | (22.6) |
| Richest | 1148 | (26.9) | 386 | (18.5) | 259 | (17.5) | 1793 | (22.9) |
| missing | 1 | | 1 | | 0 | | 2 | |

*Restricted to ever-breastfed infants aged at least 6 months not yet weaned/with age at weaning (n=7230).
†Restricted to ever-breastfed infants aged at least 24 months still breastfed/with age at cessation of breastfeeding (figure 3) (n=5382).

to 5.42; model 3, table 3). The OR for medium urbanicity was slightly increased, though not statistically significant at p<0.05 (OR 1.45, 95% CI 0.71 to 2.96; model 4), and there was evidence of a linear trend (p value 0.008). Additional adjustment for socioeconomic variables resulted in a slight reduction in the ORs (high urbanicity OR 2.64, 95% CI 1.29 to 5.42; medium urbanicity OR 1.35, 95% CI 0.66 to 2.79; model 4).

When compared with children of mothers with no formal schooling, children of mothers with secondary education were at significantly higher odds of breastfeeding discontinuation after adjustment for all demographic and socioeconomic factors (OR 1.63, 95% CI 1.23 to 2.16; model 2, table 3). Maternal employment was associated with a slight reduction in the odds of breastfeeding discontinuation before 24 months (OR 0.77, 95% CI 0.60 to 0.99). There was no evidence that SLI quintile was independently associated with breastfeeding discontinuation by 24 months. The inclusion of urbanicity in the model did not alter the socioeconomic estimates of effect.

### Interaction effects
There was no evidence of interaction between urbanicity and mother-level socioeconomic factors (household SLI and maternal education).

### Random effects
The random effects parameters for models investigating discontinuation of breastfeeding before 24 months are presented in table 3. In the null model (model 1), the proportion of residual variance attributable to mothers (level 2, 53.7%) was much higher than the variance attributable to villages (level 3, 8.5%). The addition of urbanicity to a model including individual and mother-level factors resulted in a decrease of 8.5% in village-level variance.

## DISCUSSION
### Summary of main findings
In this study, approximately two in three children were exclusively breastfed to 6 months and a similar proportion breastfed to 24 months. At the village level, high urbanicity was associated with breastfeeding discontinuation before 24 months, but there was no evidence that urbanicity was associated with early termination of EBF. At the mother level, increased maternal education was independently associated with both indicators of suboptimal breastfeeding and high SLI associated with an increased odds of EBF for less than 6 months. Maternal employment showed a variable association with breastfeeding. The residual variation in breastfeeding outcomes suggested greater heterogeneity within villages than between villages.

### Consistency with previous studies
Our estimates of breastfeeding prevalence are largely consistent with those derived from other population-based

**Table 2** Results of multilevel logistic regression models for the association between urbanicity or individual/household socioeconomic factors and termination of exclusive breastfeeding <6 months, among ever-breastfed children under six in the APCAPS household survey*.

**Termination of exclusive breastfeeding <6 months (n=7100)**

| | | n | (%) | Unadjusted OR | (95% CI) | Model 1 (null) OR | (95% CI) | Model 2 (L1 confounders +L2SES) OR | (95% CI) | Model 3 (L1 confounders +L3urbanicity) OR | (95% CI) | Model 4 (L1 confounders +L2 SES +L3urbanicity) OR | (95% CI) |
|---|---|---|---|---|---|---|---|---|---|---|---|---|---|
| **Fixed effects** | | | | | | | | | | | | | |
| Maternal education | No formal schooling | 615 | (29.1) | ref | | | | ref | | | | ref | |
| | Primary | 534 | (38.6) | 3.42 | (2.23 to 5.25) | | | 3.37 | (2.13 to 5.31) | | | 3.36 | (2.13 to 5.30) |
| | Secondary+ | 1231 | (38.2) | 1.82 | (1.29 to 2.57) | | | 1.69 | (1.12 to 2.54) | | | 1.69 | (1.12 to 2.54) |
| Maternal employment | Not working | 1584 | (33.5) | ref | | | | ref | | | | ref | |
| | Working | 796 | (33.7) | 0.94 | (0.69 to 1.28) | | | 1.43 | (1.00 to 2.03) | | | 1.43 | (1.00 to 2.04) |
| SLI | Poorest | 280 | (26.8) | ref | | | | ref | | | | ref | |
| | Poorer | 432 | (32.9) | 1.80 | (1.07 to 3.02) | | | 1.65 | (0.97 to 2.81) | | | 1.66 | (0.98 to 2.82) |
| | Middle | 525 | (35.1) | 2.24 | (1.36 to 3.71) | | | 2.08 | (1.24 to 3.50) | | | 2.09 | (1.24 to 3.51) |
| | Richer | 564 | (35.1) | 2.19 | (1.33 to 3.59) | | | 1.98 | (1.17 to 3.34) | | | 1.99 | (1.18 to 3.35) |
| | Richest | 579 | (35.3) | 2.22 | (1.35 to 3.65) | | | 2.11 | (1.22 to 3.63) | | | 2.11 | (1.23 to 3.64) |
| | trend† (p value) | | | 0.004 | | | | 0.003 | | | | 0.015 | |
| Urbanicity | Low | 459 | (33.7) | ref | | | | | | ref | | ref | |
| | Medium | 517 | (27.5) | 0.49 | (0.15 to 1.56) | | | | | 0.48 | (0.15 to 1.54) | 0.48 | (0.15 to 1.51) |
| | High | 1404 | (36.4) | 1.10 | (0.35 to 3.45) | | | | | 1.09 | (0.34 to 3.43) | 1.04 | (0.34 to 3.20) |
| | trend† (p value) | | | 0.87 | | | | | | 0.89 | | 0.91 | |
| **Random effects** | | | | | | | | | | | | | |
| Level 2 (mothers) | variance (SE) | | | | | 13.619 | (1.2202) | 13.9036 | (1.2454) | 13.9888 | (1.2531) | 13.9072 | 1.2456 |
| | PCV (compared with null)(%)‡ | | | | | ref | | −2.09 | | −2.71 | | −2.11 | |
| Level 3 (villages) | variance (SE) | | | | | 1.4949 | (0.4712) | 1.4429 | (0.4574) | 1.3870 | (0.4427) | 1.3200 | 0.4235 |
| | PCV (compared with null)(%)‡ | | | | | ref | | 3.48 | | 7.22 | | 11.70 | |
| | PCV (compared with model 2)(%)§ | | | | | – | | ref | | – | | 8.52 | |

*All ORs calculated using multilevel modelling and complete case sample (see figure 2).
†Test for trend: p value including variable as linear.
‡Proportional change in variance ((model 1 variance – model X variance)/model 1 variance)*100%.
§Proportional change in variance ((model 2 variance – model 4 variance)/model 2 variance)*100%.
APCAPS, Andhra Pradesh Children and Parents Study; SLI, standard of living index.

**Table 3** Results of multilevel logistic regression models for the association between urbanicity or individual/household socioeconomic factors and discontinuation of continued breastfeeding <24 months, among ever-breastfed children under six in the APCAPS household survey*

**Discontinuation of breastfeeding <24 months (n=5288)**

| | | n | (%) | Unadjusted | | Model 1 (null) | | Model 2 (L1 confounders +L2 SES) | | Model 3 (L1 confounders +L3 urbanicity) | | Model 4 (L1 confounders +L2 SES +L3 urbanicity) | |
|---|---|---|---|---|---|---|---|---|---|---|---|---|---|
| | | | | OR | (95% CI) | OR | (95% CI) | OR | (95% CI) | OR | (95% CI) | OR | (95% CI) |
| **Fixed effects** | | | | | | | | | | | | | |
| Maternal education | No formal schooling | 521 | (30.7) | ref | | | | ref | | | | ref | |
| | Primary | 395 | (37.3) | 1.48 | (1.13 to 1.94) | | | 1.21 | (0.88 to 1.65) | | | 1.21 | (0.88 to 1.65) |
| | Secondary+ | 1079 | (42.5) | 2.28 | (1.82 to 2.87) | | | 1.63 | (1.23 to 2.16) | | | 1.62 | (1.22 to 2.15) |
| Maternal employment | Not working | 1384 | (41.3) | ref | | | | ref | | | | ref | |
| | Working | 611 | (31.6) | 0.55 | (0.45 to 0.67) | | | 0.77 | (0.60 to 0.99) | | | 0.78 | (0.61 to 0.99) |
| SLI | Poorest | 271 | (34.6) | ref | | | | ref | | | | ref | |
| | Poorer | 370 | (37.3) | 1.38 | (0.99 to 1.91) | | | 1.18 | (0.82 to 1.69) | | | 1.17 | (0.82 to 1.68) |
| | Middle | 374 | (33.1) | 1.07 | (0.78 to 1.48) | | | 0.86 | (0.60 to 1.24) | | | 0.86 | (0.60 to 1.23) |
| | Richer | 467 | (39.4) | 1.5 | (1.09 to 2.06) | | | 1.05 | (0.73 to 1.51) | | | 1.04 | (0.73 to 1.50) |
| | Richest | 513 | (42.9) | 1.75 | (1.28 to 2.41) | | | 1.09 | (0.74 to 1.59) | | | 1.08 | (0.74 to 1.57) |
| | Trend†(p value) | | | 0.001 | | | | 0.027 | | | | 0.915 | |
| Urbanicity | Low | 290 | (29.5) | ref | | | | | | ref | | ref | |
| | Medium | 494 | (34.6) | 1.40 | (0.72 to 2.71) | | | | | 1.45 | (0.71 to 2.96) | 1.35 | (0.66 to 2.79) |
| | High | 1213 | (42.2) | 2.74 | (1.42 to 5.28) | | | | | 2.96 | (1.45 to 6.05) | 2.64 | (1.29 to 5.42) |
| | Trend† (p value) | | | 0.003 | | | | | | 0.003 | | 0.008 | |
| **Random effects** | | | | | | | | | | | | | |
| Level 2 (mothers) | Variance (SE) | | | | | 3.2070 | (0.4847) | 4.1838 | (0.6347) | 4.2895 | (0.6445) | 4.1793 | (0.6342) |
| | PCV (compared with null)(%)‡ | | | | | ref | | −30.46 | | −33.75 | | −30.32 | |
| Level 3 (villages) | Variance (SE) | | | | | 0.6036 | (0.1966) | 0.6677 | (0.2221) | 0.4942 | (0.1718) | 0.5029 | (0.1739) |
| | PCV (compared with null)(%)‡ | | | | | ref | | −10.62 | | 18.12 | | 16.68 | |
| | PCV (compared with model 2)(%)‡ | | | | | – | | ref | | – | | 24.68 | |

*All ORs calculated using multilevel modelling and complete case sample (see figure 2).
†Test for trend: p value including variable as linear.
‡Proportional change in variance ((model 1 variance–model X variance)/model one variance)*100%.
§Proportional change in variance ((model 2 variance – model 4 variance)/model 2 variance)*100%.
APCAPS, Andhra Pradesh Children and Parents Study; SLI, standard of living index.

studies in India. Early results from National Family Health Survey (NFHS-4) (2015–2016) Telangana state indicate that 67.3% of infants aged 0–6 months (at the time of survey) were exclusively breastfed,[19] and a study of 600 mother–child pairs in Andhra Pradesh reports that 75% of infants aged 3–5 months were exclusively breastfed.[20] Some of the younger infants included in these two study samples will have ceased breastfeeding by 6 months, suggesting that our study sample has a slightly higher proportion of EBF to 6 months. The overall proportion of children breastfed until at least 24 months in our study was almost identical to an analysis of all-India NFHS-2 data: (62.2% vs 63%).[6]

Very few existing studies have investigated the association between urbanicity and breastfeeding. In one study based in the Philippines, Dahly and Adair reported that length of breastfeeding was negatively correlated with increasing urbanicity (using a multicomponent measure).[13] The persisting association between high urbanicity and increased odds of breastfeeding discontinuation <24 months—after adjustment for lower level socioeconomic circumstances—reported in our study support the findings from Dahly and Adair.

Increasing urbanicity is associated with positive socioeconomic changes such as improved education for women and increased income and household wealth. A number of other studies from India and other LMICs have demonstrated a negative association between improved socioeconomic position and breastfeeding practices.[6–8 21–24] We found similar results with regard to household SLI and increased maternal education, and early termination of EBF. One explanation for this trend could be the greater affordability and/or social desirability of commercial breastmilk substitutes. The association between education and early termination of EBF was strongest for primary education (primary education: OR 3.37, 95% CI 2.13 to 5.31; secondary education: OR 1.69, 95% CI 1.12 to 2.54). This suggests that while education in general is associated with a reduction in the length of EBF, higher levels of education partially ameliorate this effect. Interestingly, there was some evidence that maternal employment had a protective effect on breastfeeding discontinuation by 24 months, though the opposite trend was observed with regard to early cessation of EBF. There is some evidence of a U-shaped association between education and women's employment in India, with paid employment outside the home common among women with little or no formal education, lower among women with moderate levels of education and rising again with high levels of education.[25] Mothers in employment are likely to be a heterogeneous group, making it difficult to draw any firm conclusions about the association between paid employment and breastfeeding practices in this sample.

## Strengths and limitations

The APCAPS cohort provides a unique opportunity to investigate current health behaviour and outcomes set against the backdrop of rapid urbanisation and economic transition in rural India. While studies based on the high level urban–rural comparison may help to predict the impact of 'total' urbanisation on breastfeeding practices, they obscure the temporal emergence of subtler changes in the urban environment that may be amenable to intervention. The use of multilevel models enabled us to explore the role of factors at different levels: individual, mother and village.

Although the vast majority of all under 6s in the study villages were included in our analysis, 15.7% (n=1464) were excluded due to missing information on feeding history due to the mother living elsewhere, travelling or deceased. In a comparison of included and excluded children, there was no evidence that infant sex, infant age, number of under 6s living in the household or household SLI differed by missing status (see online supplementary table 1). A slightly higher proportion of excluded children resided in high urbanicity villages (p 0.09).

We relied on maternal recall of breastfeeding events for our outcome measurement. For the analysis of EBF at 6 months, the recall period ranged from 0 to 5.5 years, and for breastfeeding continuation at 24 months the recall period was 0–4 years. A review of 11 studies assessing the validity and reliability of maternal recall of breastfeeding concluded that maternal recall of breastfeeding duration is good, especially when the recall period is short (<3 years).[26] A more recent study, conducted in a population where breastfeeding initiation was near universal and duration long, found that even after 20 years, 64% of women recalled duration correctly to within 1 month (90% within 3 months).[27] However, there is some evidence that recall of age at introduction of complementary foods or non-breastmilk fluids is less accurate.[26] It is unclear whether any misclassification of breastfeeding behaviour is independent of other characteristics, but where differential misclassification has been suggested, more highly educated or wealthier mothers have tended to over-report breastfeeding.[28] Given that these characteristics were associated with suboptimal breastfeeding practices in this study, we may have underestimated any true difference in breastfeeding by sociodemographic characteristics.

Our measure of urbanicity was derived from NLTI data, information that is objective, regularly updated and free to use. Additionally, data on NLTI are available over a number of years and could be used in future studies to investigate trends in urbanicity over time. However, it must be noted that urbanicity is an ecological indicator and as such may not accurately reflect individual environment, particularly given that many women may travel regularly outside their home village for work or family reasons.

## Implications

Nearly a quarter (24%) of all global under-five deaths occur in India.[29] In light of the failure to achieve the Millennium Development Goal infant mortality rate (IMR) target reduction,[30] a new target of reducing the

IMR to 20 per 1000 live births by 2020 has recently been proposed.[31] Early results from the latest NFHS-4 data collected in Telegana state report a current IMR of 28 (20 in urban areas, 35 in rural areas).[19] An increase in optimal breastfeeding practices will help to achieve improvements in infant survival, in addition to reducing the considerable burden of infant morbidity.[3 32] India faces an ever-increasing epidemic of chronic disease in common with many other LMICs. Several studies have suggested that breastfeeding has a protective effect on long-term outcomes such as obesity and diabetes in adulthood,[33] though residual confounding is difficult to exclude,[34] and the most recent data from the PROBIT RCT do not support an association between breastfeeding and adiposity in late childhood.[35]

A substantial proportion of infants in India are exclusively breastfed for less than the 6 months recommended by WHO,[7 9 10] and a recent study reported that there was little change in the prevalence of EBF in India between 1992–1993 and 2005–2006[8] . The lack of country-specific holistic and coordinated policy programmes supporting breastfeeding has also been highlighted.[36] Therefore, research to further understand the determinants of suboptimal breastfeeding practices in India is timely.

Our findings suggest that in LMICs with a strong tradition of breastfeeding, negative changes in breastfeeding behaviour may be observed during early stages of the urbanicity transition. Reduced duration of breastfeeding among more educated mothers may be one of the earliest markers of this change. India is currently undergoing rapid urbanisation, with the proportion of the population living in towns and cities is set to increase from an estimated 28% in 2011 to 38% by 2026[37]. Many more individuals live in areas which though traditionally described as rural are increasingly displaying many of the characteristics of urban areas. There is good evidence that breastfeeding behaviours are amenable to change through interventions delivered at the household and community level, as well as those targeting health systems.[38 39] Intervention programmes to protect and promote breastfeeding should be considered in transitioning communities to counteract changes in breastfeeding practices, preferably targeted at those mothers identified as most at risk of suboptimal breastfeeding practices.

**Author affiliations**
[1]Department of Non-communicable Disease Epidemiology, London School of Hygiene, London, UK
[2]Indian Institute of Public Health, Hyderabad, India
[3]Department of Anthropology, University of Delhi, New Delhi, India
[4]Centre for Control of Chronic Conditions, Public Health Foundation of India, New Delhi, India
[5]National Institute of Nutrition, Hyderabad, India
[6]Department of Geography and Environment, University of Southampton, Southampton, UK
[7]School of Social and Community Medicine, University of Bristol, Bristol, UK

**Acknowledgements** We wish to acknowledge our dedicated field teams led by Santhi Bhogadi and the study participants who made this study possible. We also acknowledge the contribution of Naveen Chittaluri and Ekta Jain to data processing and management. We also thank Cono Ariti at the London School of Hygiene and Tropical Medicine (LSHTM) who provided statistical advice, and Poppy Mallinson (LSHTM) for assistance with calculating the SLI.

**Contributors** The study was conceived and designed by Shah Ebrahim (SE) and VG, and overall study management was by VG, GKW, KVRK and SK. Study tools were developed by VG, GKW and AA, and the study implemented by VG and GKW. Data management was provided by AA, GKW, SB and CB. SB was in charge of field management, SA and CB contributed to data collection and processing. RTW obtained and processed the NTLI data. LO, CB, SA and SK designed the analysis reported here. LO performed the statistical analysis, SK helped interpret the results and provided crucial input on manuscript preparation. LO, CB and SK were responsible for the initial draft of the manuscript. All authors contributed to the revision of the manuscript and reviewed and approved the final version.

**Funding** The APCAPS household survey was funded by a Wellcome Trust Strategic Award (Grant: 084674/Z, principal investigator Shah Ebrahim).

**Competing interests** None declared.

**Patient consent** Not needed.

**Ethics approval** Ethics committees at the National Institute of Nutrition (Hyderabad, India) and London School of Hygiene and Tropical Medicine (London, UK).

**Provenance and peer review** Not commissioned; externally peer reviewed.

**Data sharing statement** For details on how to access APCAPS data, please visit http://apcaps.lshtm.ac.uk.

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
