## [Reviewer comments · BMJ Open]

ARTICLE DETAILS

TITLE (PROVISIONAL)	Is increasing urbanicity associated with changes in breastfeeding duration in rural India? An analysis of cross-sectional household data from the Andhra Pradesh Children and Parents Study
AUTHORS	Oakley, Laura; Baker, Chris; Addanki, Srivalli; Gupta, Vipin; Walia, Gagandeep; Aggarwal, Aastha; Bhogadi, Santhi; Kulkarni, Bharati; Wilson, Robin; Prabhakaran, Dorairaj; Ben-Shlomo, Yoav; Davey Smith, George; Radha Krishna, KV; Kinra, Sanjay

VERSION 1 - REVIEW

REVIEWER	Tuan Nguyen FHI 360, Vietnam
REVIEW RETURNED	08-Apr-2017

GENERAL COMMENTS	This paper is interesting and has a potential. Overall, this work seems to make an important contribution to examining the association between urbanity and socio-economic factors breastfeeding practices in rural communities in India. This manuscript could benefit from additional information provided in the methods, results, and discussion sections. 1. My key concern is about the study outcomes. - The two outcomes were not clearly defined (pg. 6). Please provide an appendix about related questions.- However, it seems to me that the authors did not use the WHO recommended indicators to evaluate infant and young child feeding practices. If so, the findings are not readily to be compared with other studies (e.g., as did in pg. 13-14) or be used by other scientists or policy makers. Please check the reference 26 for WHO's definitions of exclusive breastfeeding < 6 months, timely introduction of complementary foods, and breastfeeding at 24 months.- The outcomes' validity might be affected by recall and rounding bias, especially for those who aged 3-5 years old. Although in the limitations, the authors mentioned about the recall bias, they considered them negligible and cited 2 references. To my knowledge, the two references indicated that the recall bias is problematic for similar cases. For example, 24-hour dietary recall suggested by WHO to minimize the recall bias.- Given the data might be collected in a continuous format for the two outcomes, please provide mean, SD, median, and range. 2. Because the study conducted in low SES villages in India, the authors should discuss clearly about the finding about urbanization and SES in this context. 3. Please provide more detail information about data collection (pg. 5). Because this is a cross-sectional data analysis of data from a
--

	longitudinal study, the claim on the strength associated with longitudinal study (pg. 14) is not warranted.
--	---

REVIEWER	Mona Nabulsi American University of Beirut Medical Center. Beirut, Lebanon.
REVIEW RETURNED	09-Apr-2017

GENERAL COMMENTS	This is a well-done cross-sectional study that is of public health relevance to India mainly. I have few comments to the authors:  1. Strengths and limitations: The authors minimized the effect of recall bias since the children were less than 6 years and recall period was "short". This statement was backed up with 2 references from LMICs. I would argue that 6 years is not a "short" period, especially for mothers with several children, with poor SES and low education. Recall bias is the major limitation of this study. 2. Originality: The risk factors investigated in this study, including "urbanicity" are well-known to be associated with shorter duration of EBF and any breastfeeding. Hence, the findings were quite predictable. The ascertainment of urbanicity using the night-time light intensity is very original. 3. The random-effects analysis is complex and I do not have the expertise required to critique this part of the statistical analysis. Good luck.
--

VERSION 1 – AUTHOR RESPONSE

Reviewer: 1

Tuan Nguyen

FHI 360, Vietnam

Please state any competing interests or state 'None declared': No

Please leave your comments for the authors below

This paper is interesting and has a potential. Overall, this work seems to make an important contribution to examining the association between urbanity and socio-economic factors breastfeeding practices in rural communities in India. This manuscript could benefit from additional information provided in the methods, results, and discussion sections.

1. My key concern is about the study outcomes.

- The two outcomes were not clearly defined (pg. 6). Please provide an appendix about related questions.

Response

We have clarified the outcomes as described on page 6 (paragraph 1 on "Breastfeeding outcomes and explanatory variables").

- However, it seems to me that the authors did not use the WHO recommended indicators to evaluate infant and young child feeding practices. If so, the findings are not readily to be compared with other studies (e.g., as did in pg. 13-14) or be used by other scientists or policy makers. Please check the reference 26 for WHO's definitions of exclusive breastfeeding < 6 months, timely introduction of complementary foods, and breastfeeding at 24 months.

Response

Thank you for this observation. We have not claimed to use the WHO recommended indicators as described in reference 26. The WHO recommended indicators are 'point-in-time' measurements.

Applying these indicators to our data would have resulted in extremely small samples. On page 6 (“paragraph 1 on “Breastfeeding outcomes and explanatory variables”) we state that our outcomes reflect failure to achieve two of the WHO recommendations for optimum feeding practices (exclusive breastfeeding to six months and continued breastfeeding to two years). We feel this is an accurate description of outcomes used in our study. On page 13 (“Consistency with previous studies”, paragraph 1) we compared our results to early results from NFHS-4, noting that the NFHS-4 denominator was infants aged 0-6 months (as per WHO IYCF indicators). We have added some additional text to make this clearer. The second comparison is with an analysis of NFHS-2 data, where the investigators report the probability of still being breastfed at 24 months, directly comparable to the equivalent outcome in our study.

- The outcomes’ validity might be affected by recall and rounding bias, especially for those who aged 3-5 years old. Although in the limitations, the authors mentioned about the recall bias, they considered them negligible and cited 2 references. To my knowledge, the two references indicated that the recall bias is problematic for similar cases. For example, 24-hour dietary recall suggested by WHO to minimize the recall bias.

Response

As the reviewer acknowledges, we accept the possibility of recall bias. We have revised and expanded our discussion of recall bias (page 15, “Strengths and Limitations”. Paragraph 3). We accept that some women will not remember the exact age at the introduction of other foods or fluids/cessation of breastfeeding, but we think that the likelihood of remembering breastfeeding behaviour at two key infant ages (six months and 24 months) is higher. We have consulted the literature on the validity and reliability of maternal recall, and the issue of whether misclassification of breastfeeding behaviour is likely to be differential or non-differential remains inconclusive. Studies which have reported that the accuracy of recall varies by sociodemographic characteristics have generally found that wealthier and more educated women tend to over-estimate breastfeeding duration. In our study, these characteristics were associated with a lower odds of optimal breastfeeding, suggesting that if anything, we may have underestimated any association between sociodemographic characteristics/urbanicity and breastfeeding practices.

- Given the data might be collected in a continuous format for the two outcomes, please provide mean, SD, median, and range.

Response

We have provided the mean, SD, median and IQR (we considered the latter more informative than the range due to extreme values).

2. Because the study conducted in low SES villages in India, the authors should discuss clearly about the finding about urbanization and SES in this context.

Response

We have interpreted our findings in the context of policy implications and increasing urbanisation in India. We are not confident we understand Reviewer 1’s point, perhaps they could clarify?

3. Please provide more detail information about data collection (pg. 5). Because this is a cross-sectional data analysis of data from a longitudinal study, the claim on the strength associated with longitudinal study (pg. 14) is not warranted.

Response

We have removed the second mention of cohort in this paragraph (“Strengths and limitations”, paragraph 1), but left in the first mention as this is being used to describe the study sample.

Reviewer: 2

Mona Nabulsi

American University of Beirut Medical Center. Beirut, Lebanon.

Please state any competing interests or state 'None declared': None declared.

Please leave your comments for the authors below

This is a well-done cross-sectional study that is of public health relevance to India mainly. I have few comments to the authors:

1. Strengths and limitations: The authors minimized the effect of recall bias since the children were less than 6 years and recall period was "short". This statement was backed up with 2 references from LMICs. I would argue that 6 years is not a "short" period, especially for mothers with several children, with poor SES and low education. Recall bias is the major limitation of this study.

Response

We have addressed this point in our response to reviewer 1's comments.

2. Originality: The risk factors investigated in this study, including "urbanicity" are well-known to be associated with shorter duration of EBF and any breastfeeding. Hence, the findings were quite predictable. The ascertainment of urbanicity using the night-time light intensity is very original.

Response

Thank you for this comment. We agree that it is well-known that socio-demographic variables such as education and wealth are associated with breastfeeding, however, we would emphasise that there have been very few studies which have looked at the independent effect of area-based measures of 'urbanicity', and breastfeeding.

3. The random-effects analysis is complex and I do not have the expertise required to critique this part of the statistical analysis.

VERSION 2 – REVIEW

REVIEWER	Mona Nabulsi, MD, MS Department of Pediatrics and Adolescent Medicine American University of Beirut Medical Center Beirut-Lebanon
REVIEW RETURNED	15-May-2017

GENERAL COMMENTS	I have no further comments to the authors as they had responded to all previous comments.
---